# Does an Endometrial Cancer Diagnosis among Asymptomatic Patients Improve Prognosis?

**DOI:** 10.3390/cancers14010115

**Published:** 2021-12-27

**Authors:** Petra Vinklerová, Petra Ovesná, Markéta Bednaříková, Luboš Minář, Michal Felsinger, Jitka Hausnerová, Vít Weinberger

**Affiliations:** 1Department of Gynecology and Obstetrics, University Hospital Brno and Faculty of Medicine, Masaryk University, 60200 Brno, Czech Republic; vinklerova.petra@fnbrno.cz (P.V.); minar.lubos@fnbrno.cz (L.M.); felsinger.michal@fnbrno.cz (M.F.); 2Institute of Biostatistics and Analyses, Faculty of Medicine, Masaryk University, 60200 Brno, Czech Republic; ovesna.petra@fnbrno.cz; 3Department of Internal Medicine, Hematology and Oncology, University Hospital Brno and Faculty of Medicine, Masaryk University, 60200 Brno, Czech Republic; bednarikova.marketa@fnbrno.cz; 4Department of Pathology, University Hospital Brno and Faculty of Medicine, Masaryk University, 60200 Brno, Czech Republic; hausnerova.jitka@fnbrno.cz

**Keywords:** endometrial cancer, postmenopausal bleeding, prognosis

## Abstract

**Simple Summary:**

Endometrial cancer is common malignancy with an excellent prognosis due to its early symptoms—abnormal bleeding. It is still common in some countries to provide a biopsy in asymptomatic patients based on ultrasound findings; even though, it is not supported by the European guidelines. The aim of our study was to find out if there is a prognostic difference among symptomatic and bleeding-free patients with similar clinical histological characteristics.

**Abstract:**

Background: Endometrial cancer is the most common gynecological malignancy in developed countries with no screening available. There is still a tendency to provide invasive bioptic verification in asymptomatic women with abnormal ultrasound findings to diagnose carcinoma in a preclinical phase; even though, it is not supported by European guidelines. Our goal was to determine DFS (disease-free survival), OS (overall survival), and DSS (disease-specific survival) differences between symptom-free and symptomatic (bleeding, or spotting) endometrial cancer patients with similar stage and tumor/clinical characteristics. Methods: All of our patients with endometrial cancer following surgical treatment between 2006 and 2019 were assessed, evaluating risk factors for recurrence and death while focusing on bleeding using univariable and multivariable analysis. Results: 625 patients meeting the inclusion criteria were divided into asymptomatic (*n* = 144, 23%) and symptomatic (*n* = 481, 77%) groups. The median follow-up was 3.6 years. Using univariable analysis, symptomatic patients had a three times higher risk of recurrence (HR 3.1 (95% Cl 1.24–7.77), *p* = 0.016). OS (HR 1.35 (0.84–2.19), *p* = 0.219) and DSS (HR 1.66 (0.64–4.28), *p* = 0.3) were slightly worse without reaching statistical significance. In our multivariable analysis, symptomatology was deemed completely insignificant in all monitored parameters (DFS: HR 2.03 (0.79–5.24), *p* = 0.144; OS: HR 0.72 (0.43–1.21), *p* = 0.216). Conclusions: The symptomatic endometrial cancer patients risk factor of earlier recurrence and death is insignificantly higher when compared with the asymptomatic cohort. However, multivariable analysis verifies that prognosis worsens with other clinically relevant parameters, not by symptomatology itself. In terms of survival outcome in EC patients, we recognized symptomatology as a non-significant marker for the patient’s prognosis.

## 1. Introduction

In developed countries, endometrial cancer (EC) is the most common gynecological malignancy with more than 380,000 new cases in 2018 [1]. It is usually diagnosed in the first stage with an excellent prognosis identifying thickened or abnormal endometrium via ultrasound use in daily practice and/or early symptoms including postmenopausal or abnormal bleeding, which occurs in 90% of cancers [2]. Among European women, the relative 5-year survival is 76% [3]. The 5-year overall survival rate is 89% with stage I, 78% stage II, 61% stage III, and 21% with stage IV [4]. Along with FIGO (The International Federation of Gynecology and Obstetrics) stage, other conventional prognostic factors are histology, grade, and lymphovascular space invasion (LVSI) [5]. Currently, these factors still determine patient risk and the indication of adjuvant treatment.

Furthermore, additional immunohistochemical markers seem to be beneficial in predicting prognosis, even though they have never been part of risk classification. Abnormal expression of L1CAM (L1 Cell Adhesion Molecule), mutated tumor protein p53, a loss of estrogen (ER), and progesterone receptors (PR), for example, are associated with a worse prognosis [6].

The latest EC classification trend is grouping according to molecular features, first introduced in 2013 by the Cancer Genome Atlas (TCGA) Research Network [7]. They divided EC into four prognostic subclasses. The polymerase-epsilon (POLE) ultramutated was characterized by a very favorable outcome. The hypermutated group (also termed microsatellite instable, MSI) and the copy-number low (microsatellite stable, MSS) were associated with intermediate results. The copy-number high (mainly serous histotype) was defined by tumor protein p53 mutations and poor prognosis. Recently published guidelines by the European Society of Gynecological Oncology (ESGO), the European Society for Radiotherapy and Oncology (ESTRO), and the European Society of Pathology (ESP), now recommend this EC classification; however, it is not yet comprehensive. Worldwide use in daily practice is hindered by its cost and availability.

General population screening is not recommended, apart from patients with Lynch syndrome [5,8]. Women should be advised to report any postmenopausal or abnormal bleeding, which is at higher risk for malignity presence. Daily routine ultrasound use may lead to incidental findings of thickened endometrium or polyps and invasive procedures (resulting from fear of malignancy). Based on a thickened endometrium ≥5 mm, it is still common in many countries to perform routine endometrial sampling in asymptomatic postmenopausal women, even though this is not supported by European guidelines [8]. Endometrial cancer prevalence among asymptomatic women is low, and biopsy is not recommended in the case of a postmenopausal patient with endometrial thickness or polyp without bleeding [9,10]. On the other hand, the rationale for an active approach could relate to a doctor’s concern about neglecting a woman’s health following late detection of uterine cancer—a real and clinically serious issue. So far, there are no robust data published regarding whether there is a difference and if it matters when endometrial cancer is diagnosed in the asymptomatic or symptomatic phase.

Our study, accordingly, aimed to answer the question of whether bleeding is a strong prognostic factor in endometrial cancer patients. We evaluated symptomatology concerning DFS (disease-free survival), OS (overall survival), and DSS (disease-specific survival). Is there an advantage when diagnosing endometrial cancer of the same stage and characteristics in the preclinical (asymptomatic) phase from the perspective of DSF, OS, and DSS?

## 2. Materials and Methods

Our retrospective observational study took place between January 2006 and December 2019 at the Department of Gynecology and Obstetrics, University Hospital Brno, Czech Republic. All surgically treated patients with EC diagnosis were consecutively included in the study. Patients were divided into two groups depending on symptoms while focusing on postmenopausal bleeding, spotting, pinkish discharge, or irregular and excessive bleeding in premenopausal women. An ultrasound finding of an endometrial tumor in premenopausal age and endometrial thickness (≥5 mm) or a polyp was a signal for biopsy among asymptomatic postmenopausal patients. We excluded cases with no surgical treatment, uterine sarcoma histology, and unknown symptomatology status.

All patients underwent a total abdominal or laparoscopic hysterectomy with bilateral salpingo-oophorectomy (ovaries were spared when patients were younger than 45 years with endometrioid histology, grade 1). Pelvic and paraaortic lymphadenectomy was performed according to actual national recommendations (only pelvic until 2013 and since that pelvic and paraaortic lymphadenectomy in high-risk carcinomas including non-endometrioid and endometrioid cancer stage ≥1B of any grading) and patient performance status [11]. We have introduced sentinel node biopsy (using indocyanine green) instead of systematic lymphadenectomy since 2019. Only women with complete remission after primary treatment were included. During the follow-up period, regular check-ups were effected every 3–4 months following primary treatment for the first two years, biannually over the next three years, and once a year thereafter. Gynecologic examination and transvaginal/transrectal plus abdominal ultrasound were obligatory. A CT (computed tomography) scan was undertaken when recurrence was suspected. We monitored all types of recurrence—local (vaginal vault), regional (pelvic structures including lymphatic nodes), and distant (extrapelvic metastasis).

Patient’s age at the time of diagnosis and relevant data regarding histological type (endometrioid, mucinous, serous, clear-cell, and carcinosarcoma), grade, LVSI, pathological stage (according to FIGO 2009), lymphadenectomy, and adjuvant treatment provided such as radiotherapy (RT), chemotherapy (CHT), and their combination (CHRT) were obtained from medical records. We divided women into five categories according to their age for univariable analysis. A continuous variable was used for multivariable analysis.

We selected DFS, OS, and DSS as prognostic parameters to compare patients in both groups. DFS is the length of time after primary cancer treatment that a patient survives without any signs or symptoms of the disease. OS is the time from either the date of diagnosis or the start of treatment for the disease patients are still alive. DSS is the time from the diagnosis date or treatment onset to the date of death from the disease. Patients who die from causes unrelated to the cancer are not counted in this measurement [12].

Patients without an event were censored upon the date of the last follow-up visit. The impact of symptomatology on DFS and OS was assessed using Cox proportional hazards model, which gives hazard ratios (HR) accompanied by a 95% confidence interval (CI). Results from univariate analysis (crude HR) were adjusted for other clinical parameters in the multivariable Cox model giving adjusted HRs. Histology, grade, and FIGO stage categories were combined in multivariate models due to low number of events in some strata. The DSS was estimated by a cumulative incidence in the presence of death from other causes than EC as a competing risk. Comparison between groups was undertaken using the Fine and Gray method. Analyses were detailed with R software (4.0.3) including survival and cmprsk (Subdistribution Analysis of Competing Risks) packages.

## 3. Results

Between 2006 and 2019, our database included 722 endometrial cancer patients of which 625 met the inclusion criteria—144 (23%) were asymptomatic and 481 (77%) reported symptoms. Table 1 summarizes clinical/histological characteristics and adjuvant treatment. Data collection and statistical analysis were detailed in March 2021, when median (interquartile range, IQR) follow-up was 3.6 years (4 days–13.8 years).

### 3.1. Disease-Free Survival

Recurrence occurred in 56 patients during the follow-up period. Five years following primary treatment, there were no signs of uterine cancer relapse among 96.1% (92.8–99.5%) asymptomatic and 86.2% (82.5–90%) symptomatic women. Using a univariable model, symptomatic cases had a three times higher risk of recurrence (HR 3.1 (95% CI 1.24–7.77), *p* = 0.016) than asymptomatic (Figure 1A). Furthermore, we observed increased recurrence risk among elderly patients, with NEC (non-endometrioid carcinoma) histology, LVSI presence, endometrioid EC grade and stage increases, following lymphadenectomy, and CHT treatment (Table 2).

However, symptomatology became non-significant in the multivariable analysis when it was adjusted with other clinical parameters (HR 2.03 (0.79–5.24), *p* = 0.144). Concurrently, histology, grade, and stage remained risky (Table 2).

### 3.2. Overall Survival

Overall, 117 patients died during the follow-up period (20 symptom-free and 97 bleeding). We recorded a worsening OS trend in bleeding patients without reaching statistical significance with the univariable model (HR 1.35 (0.84–2.19), *p* = 0.219; Figure 1B). Five-year OS was 82.9% (75.5–90.9%) in asymptomatic and 80.1% (76–84.5%) in symptomatic cases. Reduced survival was notable among women over age 70, after lymphadenectomy, following CHT treatment, with LVSI, with a grade 3 endometrioid or NEC histology, and advancing disease stage (Table 3).

Nonetheless, symptomatology was recognized in the multivariable model with slightly longer survival in the symptomatic group (HR 0.72 (0.43–1.21), *p* = 0.216) when it was adjusted for other parameters. We recorded shorter survival in patients with grade 3 endometrioid or NEC histology and with disease stages increasing following adjustment. Adjuvant therapy had a protective effect (radiotherapy *p* = 0.067, chemoradiotherapy, *p* = 0.024) as well (Table 3).

### 3.3. Disease-Specific Survival

Thirty-six deaths were caused by endometrial cancer. Disease-specific survival was insignificantly worse with bleeding patients in the univariable model (HR 1.66 (0.64–4.28), *p* = 0.3, Figure 1C). Survival rates were substantially reduced following lymphadenectomy, CHT, with LVSI and disease stage escalation (Table 4). NEC histology and endometrioid EC grade >1 was also considered a risk, since all fatalities came from these groups of patients. The multivariate model of DSS was not performed due to the low number of events.

## 4. Discussion

Endometrial cancer is a common malignancy with a generally favorable prognosis. There is no recommended screening for the general population; however, ultrasound use in daily practice may lead to fortuitous findings of uterine polyps or hyperplasia. Since it is not possible to provide biopsies for all patients, there is a clinically driven need to establish a cut-off for identifying high-risk EC patients. A 12% prevalence of thickened endometrium ≥5 mm in gynecologically healthy asymptomatic postmenopausal women was identified with a Swedish population study [13]. A reasonable endometrial thickness threshold seems to be ≥11 mm for biopsy in asymptomatic patients when the EC incidence probability is about 6.7% compared to 1.7% in women with endometrium thickness <11 mm [10,14]. We omitted the division of asymptomatic patients according to endometrial thickness in our current study. However, we postulated before, that significant risk of malignancy is only when threshold of 12 mm was used (OR 3.54, *p* = 0.024) comparing to 8 and 5 mm [15].

Malignancy risk is less than 2% with asymptomatic polyps, and small ones can even vanish spontaneously in premenopausal women [9]. An invasive approach should be reserved for bleeding patients and in cases of infertility.

A hysteroscopy (instead of dilatation and curettage) is recommended to obtain a representative sample or remove a focal lesion, although complication risk (uterine perforation, bowel damage, bleeding, infection, fluid-overload syndrome, etc.) is not negligible [9,13,16]. Scrimin and al. published their study of 1070 patients undergoing hysteroscopy, where nearly half of the indication was inappropriate [17]. We should consider the consequences of each invasive procedure to avoid unnecessary overtreatment and potential adverse events. Even when strictly respecting the 11 mm threshold for symptom-free postmenopausal women, 19 redundant endometrial biopsies have to be undertaken to diagnose one endometrial cancer or precancerosis [10].

In our previous study, we took a closer look at the exact description of symptom-free and symptomatic endometrial cancer tumors. Asymptomatic tumors were more often endometrioid grade 1 (41.7%) compared to bleeding (17.9%), which were more frequently endometrioid grade 3 (14.6 vs. 3.5%) or NEC (11.4 vs. 6.2%). Although immunohistochemical markers L1CAM, p53, ER, PR are strongly associated with patient prognosis and survival, we did not find any significant difference in their expression between symptomatic and symptom-free EC patients. A deep myometrial and/or cervical invasion was more commonly observed in symptomatic cases. The bleeding may correspond more with the local status of the spread and very probably has no connection to the EC patient’s survival outcome [18].

Only a few studies have focused on symptomatology in EC with the view of survival. Gemer et al. presented the largest retrospective multicentric study of 1607 postmenopausal women and detailed no difference between asymptomatic and bleeding EC patients in terms of 5-year recurrence-free survival (79.1% vs. 79.4%, *p* = 0.85), disease-specific survival (83.2% vs. 82.2%, *p* = 0.57), and overall survival (79.7% vs. 76.8%, *p* = 0.37) using univariable analysis [19]. Interestingly, they did not recognize a difference in low and high-grade histology between groups; however, there was a lower deep myometrial invasion rate in symptom-free patients. Comparing our results, we affirmed that bleeding patients had a three times higher risk of recurrence (HR 3.1 (1.24–7.77), *p* = 0.016), while overall and disease-specific survival was insignificantly worse (HR 1.35 (0.84–2.19), *p* = 0.219; HR 1.66 (0.64–4.28), *p* = 0.300). Nonetheless, when using multivariable analysis symptomatology became insignificant replaced by other factors which worsened patient prognosis (LVSI, histology, grade, FIGO stage). We confirmed factors that improve patient prognosis including radiotherapy and chemoradiotherapy in endometrial cancer patients.

Similar prognostic factors were identified in another multicentric study of 543 postmenopausal women [20]. Seebacher et al. demonstrated that tumor stage, grade, patients’ age—but not symptomatology—were associated with disease-free (HR 0.9, *p* = 0.7) and overall survival (HR 0.8, *p* = 0.4) in multivariable analysis. Their conclusions are compatible with our results: Symptomatology was not a significant risk factor when a multivariable analysis was used (DFS—HR 2.03 (0.79–5.24), *p* = 0.144; OS—HR 0.72 (0.431–1.21), *p* = 0.216).

The Israeli authors divided EC patients into three groups: Asymptomatic, bleeding up to 3 months, and bleeding more than 3 months. They presented consistent results regarding deep myometrial invasion in stage I (21%, 24%, 26%, *p* = 0.84), grade 3 tumors (10%, 13%, 14%, *p* = 0.42), and advanced-stage disease (12%, 14%, 15%, *p* = 0.92) in 220 endometrioid EC patients. The only non-significant trend toward better survival in the asymptomatic and short-term bleeding group was reported (*p* = 0.172) using univariable analysis [21]. In our study, we were unable to subdivide patients according to symptom duration owing to a lack of that specific information in medical records.

The most recent study about symptomatology as a prognostic factor is concerning only patients with preoperative suspicion of the endometrial polyp [22]. This means in majority only patients in the early stage with no signs of advanced disease. Authors find no difference in survival rates and recommended follow-up instead of biopsy in asymptomatic women. In our study we included all patients after surgical treatment, so the cohort differs and we have more advanced diseases especially in symptomatic group.

In our study, we observed that the non-bleeding group differed significantly (*p* = <0.001) in stage IA (81%) compared to the symptomatic (57%). Although the bleeding EC patients were diagnosed at the higher stage according to FIGO, there was no difference between the patients in terms of specific survival and overall survival even when using univariable analysis. Bleeding and spotting alone are not significant markers that worsen the patient’s prognosis. In terms of the diagnostic, bleeding is just one of the markers, which may, in particular cases, lead to the shift towards earlier stage detection.

To the best of our knowledge, our study represents the largest unicentric cohort dealing with symptomatology as a prognostic factor in endometrial cancer. There was an earlier recurrence and death (resulting from EC or other reasons) in bleeding patients. However, a poorer prognosis is related to other clinical and histological features, not the symptomatology itself. DFS, OS, and DSS were not worse among symptomatic patients at a similar disease stage.

Our study’s strength is reflected in the significant number of patients with guaranteed consistent treatment decisions and high-quality follow-up data. The retrospective design, and the absence of selective detail such as symptom duration, might be considered a shortcoming of sorts.

Consequently, we should educate our patients to immediately report postmenopausal or irregular bleeding and to make arrangements for dilatation and curettage or a hysteroscopy when necessary. Since there is no prognostic advantage in detecting EC in the preclinical asymptomatic phase, we recommend an expectation approach and consider the necessity of invasive biopsy in terms of possible complications and comorbidities among elderly patients.

## 5. Conclusions

Symptomatic endometrial cancer patients are at higher risk of earlier recurrence and death (both from EC and other terminal conditions, with an insignificant difference compared to the asymptomatic cohort). However, a worse prognosis resulted from other specific clinically relevant parameters, not from the bleeding itself. DSF, OS, and DSS are similar in patients at the same disease stage irrespective of symptomatology. The bleeding is not the marker worsening the prognosis. Nonetheless, EC diagnosis in the asymptomatic phase would lead to earlier stage detection. In the clinical practice, the decision regarding biopsy should be based on symptomatology and/or a significant change in the finding on the imaging method.

## Figures and Tables

**Figure 1 cancers-14-00115-f001:**
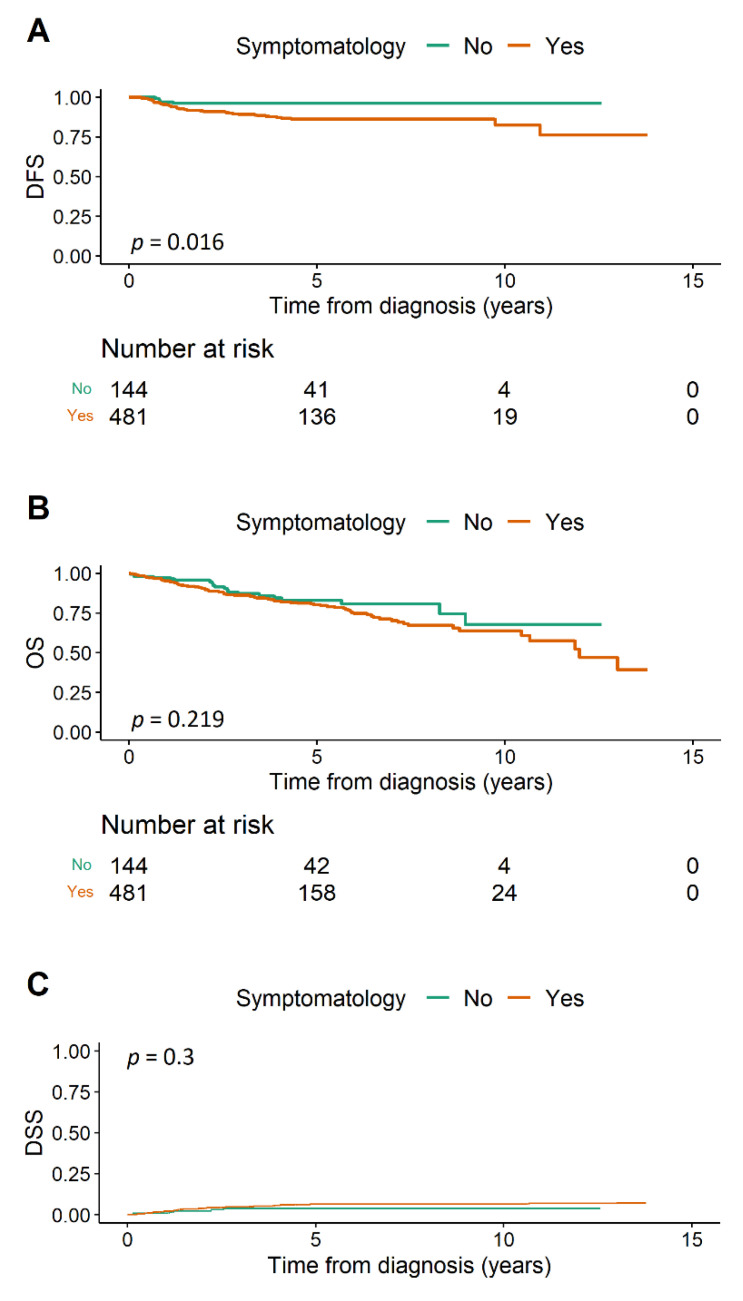
(**A**) Disease-free survival, (**B**) overall survival, (**C**) disease-specific survival. DFS = disease-free survival, OS = overall survival, DSS = disease-specific survival.

**Table 1 cancers-14-00115-t001:** Clinical patients’ characteristics.

Clinical Characteristics		Asymptomatic (*n* = 144)	Symptomatic (*n* = 481)	*p*-Value
Age (years)	<50	13 (9.0%)	46 (9.6%)	0.32
	51–60	27 (18.8%)	95 (19.8%)	
	61–70	68 (47.2%)	191 (39.7%)	
	71–80	32 (22.2%)	117 (24.3%)	
	>80	4 (2.8%)	32 (6.7%)	
Age (years)	Mean (SD)	64.5 (9.3%)	65.2 (10.6%)	0.476
Lymphadenectomy	No	124 (86.1%)	335 (69.6%)	<0.001
	Yes	20 (13.9%)	146 (30.4%)	
Adjuvant therapy	None	111 (77.6%)	264 (56.4%)	<0.001
	RT	25 (17.5%)	166 (35.5%)	
	CHT	4 (2.8%)	19 (4.1%)	
	CHRT	3 (2.1%)	19 (4.1%)	
LVSI	No	138 (95.8%)	398 (83.3%)	<0.001
	Yes	6 (4.2%)	80 (16.7%)	
Histology + grade	Endometrioid G1	60 (41.7%)	86 (17.9%)	<0.001
	Endometrioid G2	70 (48.6%)	270 (56.1%)	
	Endometrioid G3	5 (3.5%)	70 (14.6%)	
	Non-endometrioid	9 (6.2%)	55 (11.4%)	
FIGO stage	Ia	117 (81.2%)	277 (57.6%)	<0.001
	Ib	13 (9.0%)	89 (18.5%)	
	II	10 (6.9%)	60 (12.5%)	
	IIIa	1 (0.7%)	12 (2.5%)	
	IIIb	2 (1.4%)	8 (1.7%)	
	IIIc	1 (0.7%)	26 (5.4%)	
	IVa	0 (0.0%)	0 (0.0%)	
	IVb	0 (0.0%)	9 (1.9%)	

RT = radiotherapy, CHT = chemotherapy, CHRT = chemoradiotherapy, LVSI = lymphovascular space invasion, G = grade, FIGO = The International Federation of Gynecology and Obstetrics.

**Table 2 cancers-14-00115-t002:** Disease-free survival—univariable and multivariable Cox proportional hazards model.

Clinical Characteristics		Crude HR (95% CI, *p*-Value)	Adjusted HR (95% CI, *p*-Value)
Symptomatology	No	1	1
	Yes	3.1 (1.24–7.77, *p* = 0.016)	2.03 (0.79–5.24, *p* = 0.144)
Age (years)	<50	1	
	51–60	2.18 (0.47–10.08, *p* = 0.320)	
	61–70	2.49 (0.58–10.58, *p* = 0.217)	
	71–80	3.11 (0.71–13.68, *p* = 0.134)	
	>80	9.91 (2.14–45.92, *p* = 0.003)	
Age (years)	Mean (SD)	1.05 (1.02–1.08, *p* = 0.002)	1.04 (1.01–1.07, *p* = 0.013)
Lymphadenectomy	No	1	
	Yes	1.75 (1.02–3, *p* = 0.042)	
Adjuvant therapy	None	1	1
	RT	1.47 (0.81–2.69, *p* = 0.209)	0.82 (0.42–1.61, *p* = 0.569)
	CHT	9.61 (4.43–20.86, *p* < 0.001)	1.68 (0.50–5.63, *p* = 0.404)
	CHRT	2.18 (0.65–7.25, *p* = 0.205)	0.36 (0.08–1.63, *p* = 0.186)
LVSI	No	1	1
	Yes	3.75 (2.09–6.73, *p* < 0.001)	1.34 (0.58–3.06, *p* = 0.494)
Histology + grade	Endometrioid G1	1	
	Endometrioid G2	3.52 (1.06–11.63, *p* = 0.039)	1 (ref. G1 + G2)
	Endometrioid G3	7.15 (1.99–25.64, *p* = 0.003)	1.70 (0.76–3.79, *p* = 0.194)
	Non-endometrioid	15.61 (4.55–53.61, *p* < 0.001)	3.20 (1.59–6.43, *p* = 0.001)
FIGO stage	Ia	1	
	Ib	2.69 (1.33–5.47, *p* = 0.006)	
	II	3 (1.36–6.63, *p* = 0.007)	
	IIIa	6.51 (2.21–19.22, *p* = 0.001)	
	IIIb	4.11 (0.55–30.82, *p* = 0.169)	
	IIIc	7.60 (3.03–19.07, *p* < 0.001)	
	IVa	NA	
	IVb	21.67 (7.32–64.17, *p* < 0.001)	
FIGO stage	I–II	1	1
	III–IV	5.37 (2.96–9.73, *p* < 0.001)	3.55 (1.40–8.96, *p* = 0.007)

RT = radiotherapy, CHT = chemotherapy, CHRT = chemoradiotherapy, LVSI = lymphovascular space invasion, G = grade, FIGO = The International Federation of Gynecology and Obstetrics.

**Table 3 cancers-14-00115-t003:** Overall survival—univariable and multivariable Cox proportional hazards model.

Clinical Characteristics		Crude HR (95% CI, *p*-Value)	Adjusted HR (95% CI, *p*-Value)
Symptomatology	No	1	1
	Yes	1.35 (0.84–2.19, *p* = 0.219)	0.72 (0.43–1.21, *p* = 0.216)
Age (years)	<50	1	
	51–60	1.4 (0.46–4.27, *p* = 0.551)	
	61–70	1.9 (0.68–5.33, *p* = 0.222)	
	71–80	4.63 (1.66–12.89, *p* = 0.003)	
	>80	7.44 (2.48–22.31, *p* < 0.001)	
Age (years)	Mean (SD)	1.07 (1.05–1.09, *p* < 0.001)	1.07 (1.05–1.10, *p* < 0.001)
Lymphadenectomy	No	1	
	Yes	1.42 (0.98–2.07, *p* = 0.066)	
Adjuvant therapy	None	1	1
	RT	0.98 (0.64–1.51, *p* = 0.938)	0.64 (0.40–1.03, *p* = 0.067)
	CHT	5.91 (3.19–10.96, *p* < 0.001)	1.16 (0.50–2.73, *p* = 0.727)
	CHRT	1.65 (0.66–4.14, *p* = 0.284)	0.28 (0.09–0.85, *p* = 0.024)
LVSI	No	1	1
	Yes	4.55 (3.06–6.75, *p* < 0.001)	2.05 (1.13–3.72, *p* = 0.018)
Histology + grade	Endometrioid G1	1	
	Endometrioid G2	1.17 (0.66–2.07, *p* = 0.584)	1 (ref. G1 + G2)
	Endometrioid G3	2.63 (1.4–4.95, *p* = 0.003)	2.05 (1.17–3.61, *p* = 0.013
	Non-endometrioid	5.43 (2.97–9.95, *p* < 0.001)	2.89 (1.77–4.72, *p* < 0.001)
FIGO stage	Ia	1	
	Ib	2.38 (1.45–3.91, *p* = 0.001)	
	II	2.54 (1.44–4.49, *p* = 0.001)	
	IIIa	3.78 (1.60–8.94, *p* = 0.003	
	IIIb	16.01 (7.06–36.3, *p* < 0.001)	
	IIIc	8.7 (4.71–16.09, *p* < 0.001)	
	IVa	NA	
	IVb	14.89 (6.58–33.72, *p* < 0.001)	
FIGO stage	I–II	1	1
	III–IV	5.69 (3.80–8.52, *p* < 0.001)	3.63 (1.93–6.85, *p* < 0.001)

RT = radiotherapy, CHT = chemotherapy, CHRT = chemoradiotherapy, LVSI = lymphovascular space invasion, G = grade, FIGO = The International Federation of Gynecology and Obstetrics.

**Table 4 cancers-14-00115-t004:** Disease-specific survival—univariable model of cumulative incidence with competing risk.

Clinical Characteristics		Crude HR (95% CI, *p*-Value)
Symptomatology	No	1
	Yes	1.66 (0.64–4.28, *p* = 0.300)
Age (years)	<50	1
	51–60	1.14 (0.22–5.89, *p* = 0.870)
	61–70	1.48 (0.34–6.37, *p* = 0.600)
	71–80	1.7 (0.37–7.74, *p* = 0.500)
	>80	3.94 (0.77–20.05, *p* = 0.099)
Age (years)	Mean (SD)	1.03 (0.99–1.07, *p* = 0.170)
Lymphadenectomy	No	1
	Yes	2.20 (1.14–4.26, *p* = 0.019)
Adjuvant therapy	None	1
	RT	1.12 (0.49–2.53, *p* = 0.790)
	CHT	11.93 (5.17–27.53, *p* < 0.001)
	CHRT	3.75 (1.1–12.74, *p* = 0.034)
LVSI	No	1
	Yes	8.08 (4.21–15.48, *p* < 0.001)
Histology + grade	Endometrioid G1 + 2	1
	Endometrioid G3	5.19 (2.22–12.13, *p* < 0.001)
	Non-endometrioid	8.74 (4.06–18.78, *p* < 0.001)
FIGO stage	I–II	1
	III–IV	10.33 (5.36–19.90, *p* < 0.001)

RT = radiotherapy, CHT = chemotherapy, CHRT = chemoradiotherapy, LVSI = lymphovascular space invasion, G = grade, FIGO = The International Federation of Gynecology and Obstetrics.

## Data Availability

Data are available upon reasonable request.

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
