# Peer review of "Does an Endometrial Cancer Diagnosis among Asymptomatic Patients Improve Prognosis?"

_cancers, 2021, doi:10.3390/cancers14010115_

Round 1

Reviewer 1 Report

The Authors provided a very interesting study on the role of symptoms on the endometrial cancer prognosis. The study is well designed and well described. However, few comments are due:

Title

I would recommend to amend the title as more straightforward, for example “Prognostic role of symptoms at diagnosis in endometrial cancer”

Introduction

The sentence “Endometrial cancer prevalence among asymptomatic women is low, and biopsy is not recommended [9, 10]” is not clear and should be modified (e.g. the case of postmenopausal woman with endometrial thickness but with no symptoms).

Methods

  • Can the Authors specify whether the patients were consecutive or not?
  • In the abstract the Authors state that they analyzed “endometrial cancer patients with identical stage and tumor/clinical characteristics” but in Table 1 there are some differences in clinical characteristics. Maybe this sentence can be modified in the abstract.

Results

  • Table 1: please add the p value in the FIGO stage
  • In the results please add the % after the raw numbers

English grammar can be improved.

Author Response

The Authors provided a very interesting study on the role of symptoms on the endometrial cancer prognosis. The study is well designed and well described. However, few comments are due: 

Title 

I would recommend to amend the title as more straightforward, for example “Prognostic role of symptoms at diagnosis in endometrial cancer” 

Thank you for your recommendation, however we decided to preserve the original title. We think it represents the aim of the study as well as your sugestion.  

Introduction 

The sentence “Endometrial cancer prevalence among asymptomatic women is low, and biopsy is not recommended [9, 10]” is not clear and should be modified (e.g. the case of postmenopausal woman with endometrial thickness but with no symptoms). 

We rewrote the sentence to make it more clear: „Endometrial cancer prevalence among asymptomatic women is low, and biopsy is not recommended in the case of a postmenopausal patient with endometrial thickness or polyp without bleeding.“ â€¯ 

Methods 

  • Can the Authors specify whether the patients were consecutive or not? 
  • In the abstract the Authors state that they analyzed “endometrial cancer patients with identical stage and tumor/clinical characteristics” but in Table 1 there are some differences in clinical characteristics. Maybe this sentence can be modified in the abstract. 

Patients were consecutive. We added that information into methods.  

We modifed the sentence in abstract to make our goal more clear using „similar“ instead of „identical“. 

Results 

  • Table 1: please add the p value in the FIGO stage 
  • In the results please add the % after the raw numbers 

We aded p-value and % into Table 1.  

English grammar can be improved. 

The manuscript was read and edited by a native speaker again.  

Reviewer 2 Report

This is a study with a large number of cohort which aims to analyze whether diagnosis of endometrial cancer among Asymptomatic patients could improve prognosis or not. However, there were also similar studies published, e.g. Namazov et al. Maturitas. 2021 Jun;148:18-23.  doi: 10.1016/j.maturitas.2021. 04.001. which was a  multi-center retrospective cohort study to compare outcomes of symptomatic and asymptomatic women with endometrial cancer and a preoperative diagnosis of an endometrial polyp. Thus, the current study is less novel. In view of the different conclusions of other studies, the topic selection has certain clinical significance. And there are some suggestions listed below:

  1. The median follow-up was 3.6 years in the current study, which means half of included patients relapsed or progressed or died within 5 years. However, as we know that EC patients, especially patients in early stage normally have a relatively longer survival. Table 1 shows that most patients whether in Asymptomatic or Symptomatic group are in Stage I with G1-G2 without LVSI. Thus, it is a little confusing that the median follow-up date is only 3.6 years. I am wondering the possible explanations for this?
  2. In my own opinion, a slightly different opinion with the authors:

Even though symptomatology is not a significant marker for the patient´s prognosis, based on the conclusions of this study “Symptomatic endometrial cancer patients risk factor of earlier recurrence and death is insignificantly higher when compared with the asymptomatic cohort”, it is far-fetched not recommend asymptomatic patients undergo invasive biopsy.

  1. It might also be interesting to make analysis of preoperative endometrial thickness in asymptomatic women. E.g. Whether endometrium biopsy performed with endometrium ≥5mm, ≥8mm or ≥11mm could improve the diagnosis of EC in the preclinical asymptomatic phase.
  2. At the end of the paper, it was stated that “Since there is no advantage of detecting EC in the preclinical asymptomatic phase”, in my own opinion, only “there is no prognosis advantage of detecting EC in the preclinical asymptomatic phase” could be made based on the current result of this study. After all, 144 asymptomatic women out of 628 women were diagnosed with EC. Furthermore, there are significant differences of clinical parameters between Asymptomatic and Symptomatic groups, and these clinical features, such as grade, LVSI etc. remained to be risk factors for survival. Thus, there might be certain advantage for of detecting EC in the preclinical asymptomatic phase. Just based on the current results of this study, it is hard to say that there is no advantage of detecting EC in asymptomatic patients.

In summary, I would say that the current study is inadequate to meet the requirements for such a journal of Cancers.

Author Response

This is a study with a large number of cohort which aims to analyze whether diagnosis of endometrial cancer among Asymptomatic patients could improve prognosis or not. However, there were also similar studies published, e.g. Namazov et al. Maturitas. 2021 Jun;148:18-23.  doi: 10.1016/j.maturitas.2021. 04.001. which was a  multi-center retrospective cohort study to compare outcomes of symptomatic and asymptomatic women with endometrial cancer and a preoperative diagnosis of an endometrial polyp. Thus, the current study is less novel. In view of the different conclusions of other studies, the topic selection has certain clinical significance. And there are some suggestions listed below: 

You are right about similar studies being published. However, that article you mentioned consisted of 635 patients from 11 centers collected in 12 years, which could be a very heterogeneous cohort. Additionally, their study includes only patients with preoperative suspicion of an endometrial polyp – so these should be only first stage patients with no signs of advanced disease, where carcinoma was detected. We included all patients with surgical treatment, so our cohort and results are a little bit different included also advanced disease mainly in the symptomatic group. We decided to add a paragraph into „discussion“ to make it clear and compare with our data.    

“The most recent study about symptomatology as a prognostic factor is concerning only patients with preoperative suspicion of the endometrial polyp. This means in majority only patients in the early stage with no signs of advanced disease. Authors find no difference in survival rates and recommended follow-up instead of biopsy in asymptomatic women. In our study we included all patients after surgical treatment, so the cohort differs and we have more advanced diseases especially in symptomatic group.” 

  1. The median follow-p was 3.6 years in the current study, which means half of included patients relapsed or progressed or died within 5 years. However, as we know that EC patients, especially patients in early stage normally have a relatively longer survival. Table 1 shows that most patients whether in Asymptomatic or Symptomatic group are in Stage I with G1-G2 without LVSI. Thus, it is a little confusing that the median follow-up date is only 3.6 years. I am wondering the possible explanations for this? 

The median follow-up was 3.6years (4 days – 13.8 years), patients were enrolled between 2006 and 2019. We used statistical analysis which enables us to include also patient who didn´t reach the five years of follow-up.It doesn´t have to mean that they died or had recurrence during the follow-up.  

  1. In my own opinion, a slightly different opinion with the authors: Even though symptomatology is not a significant marker for the patient´s prognosis, based on the conclusions of this study “Symptomatic endometrial cancer patients risk factor of earlier recurrence and death is insignificantly higher when compared with the asymptomatic cohort”, it is far-fetched not recommend asymptomatic patients undergo invasive biopsy. 

We refined the conclusion little bit to make the recommendation less strong. 

  1. It might also be interesting to make analysis of preoperative endometrial thickness in asymptomatic women. E.g. Whether endometrium biopsy performed with endometrium ≥5mm, ≥8mm or ≥11mm could improve the diagnosis of EC in the preclinical asymptomatic phase. 

You are right, however, It has been proved several times, that the optimal threshold for asymptomatic patients is ≥11mm. We also have already published that in one of our previous study (Vinklerová P, Felsinger M, Frydová S, Ovesná P, Hausnerová J, Weinberger V. Is the finding of endometrial hyperplasia or corporal polyp an mandatory indication for biopsy? Ceska Gynekol. 2020 Winter;85(2):84-93. English. PMID:32527101.), we added that information into „Discussion“.  

“We omitted the division of asymptomatic patients according to endometrial thickness in our current study. However, we postulated before, that significant risk of malignancy is only when threshold of 12 mm was used (OR 3.54, p=0,024) comparing to 8 and 5 mm.” 

  1. At the end of the paper, it was stated that “Since there is no advantage of detecting EC in the preclinical asymptomatic phase”, in my own opinion, only “there is no prognosis advantage of detecting EC in the preclinical asymptomatic phase” could be made based on the current result of this study. After all, 144 asymptomatic women out of 628 women were diagnosed with EC. Furthermore, there are significant differences of clinical parameters between Asymptomatic and Symptomatic groups, and these clinical features, such as grade, LVSI etc. remained to be risk factors for survival. Thus, there might be certain advantage for of detecting EC in the preclinical asymptomatic phase. Just based on the current results of this study, it is hard to say that there is no advantage of detecting EC in asymptomatic patients. 

That is right, thank you for the detailed reading. We edited the sentence to make it clear, that there is no „prognostic advantage“. It is obvious that we can detect cancer in earlier stages and we admit that in „Discussion“ and „Conclusions“.